# The BIN1 rs744373 SNP is associated with increased tau-PET levels and impaired memory

Nicolai Franzmeier[1], Anna Rubinski[1], Julia Neitzel[1], Michael Ewers[1] & the Alzheimer's Disease Neuroimaging Initiative (ADNI)

The single nucleotide polymorphism (SNP) rs744373 in the bridging integrator-1 gene (BIN1) is a risk factor for Alzheimer's disease (AD). In the brain, BIN1 is involved in endocytosis and sustaining cytoskeleton integrity. Post-mortem and in vitro studies suggest that BIN1-associated AD risk is mediated by increased tau pathology but whether rs744373 is associated with increased tau pathology in vivo is unknown. Here we find in 89 older individuals without dementia, that BIN1 rs744373 risk-allele carriers show higher AV1451 tau-PET across brain regions corresponding to Braak stages II–VI. In contrast, the BIN1 rs744373 SNP was not associated with AV45 amyloid-PET uptake. Furthermore, the rs744373 risk-allele was associated with worse memory performance, mediated by increased global tau levels. Together, our findings suggest that the BIN1 rs744373 SNP is associated with increased tau but not beta-amyloid pathology, suggesting that alterations in BIN1 may contribute to memory deficits via increased tau pathology.

---

[1] Institute for Stroke and Dementia Research, Klinikum der Universität München, Ludwig-Maximilians-Universität LMU, Feodor-Lynen Straße 17, 81377 Munich, Germany. A full list of consortium members appears at the end of the paper.  Correspondence and requests for materials should be addressed to M.E. (email: michael.ewers@med.uni-muenchen.de)

Alzheimer's disease (AD) is the most common cause of late onset dementia and is characterized by the development of pathological amyloid plaques and tau tangles in the brain[1]. While late onset AD is an age-related disease, results from twin and family studies have emphasized that ~50% of phenotypic variance in AD can be explained by genetic variations[2]. Recent genome-wide association studies (GWAS) have identified several loci that are associated with increased risk of AD, among which the single nucleotide polymorphisms (SNPs) in the bridging integrator 1 (BIN1) gene show the second highest odds-ratios for sporadic AD, superseded only by apolipoprotein E (APOE) variants[3–7]. Specifically, the most frequently reported BIN1 AD risk variant is the SNP rs744373 which shows a global allele frequency of 37% and is associated with an increase in AD risk by an odds-ratio of 1.17–1.19[5,7–10]. Thus, understanding the mechanisms by which BIN1 and in particular the rs744373 SNP contributes to AD risk will lead to a better understanding of the pathomechanisms of AD and help uncover novel therapeutic targets.

The BIN1 gene encodes the nucleocytoplasmic adaptor protein BIN1 also known as amphiphysin-2, a membrane deforming protein that is most highly expressed in muscle and brain tissue in an isoform dependent way[11,12]. In the brain, BIN1 subserves multiple functions such as regulating endocytosis, cytoskeleton integrity, and apoptosis[13]. A key hypothesis for the role of BIN1 in AD is the aggravation of tau pathology, i.e. a key primary brain pathology associated with cognitive impairment in AD[14,15]. In post-mortem studies in AD, higher brain BIN1 expression was found to be associated with the presence of neurofibrillary tau tangles[16]. Importantly, previous studies have described higher BIN1 mRNA expression in brain tissue of BIN1 risk SNP carriers with and without AD[15,17]. Further, AD patients carrying a BIN1 risk SNP showed higher post-mortem tau pathology but not Aβ when compared to non-carrier AD patients[15]. Together these findings suggest that BIN1 SNP-associated alterations in BIN1 expression contribute to the development of tau pathology. Although the mechanisms that link BIN1 to tau pathology are only poorly understood, recent findings suggest that alterations in the protein level of a neuron-specific BIN1 isoform that binds exclusively to clathrin[18], enhance transmission of tau between neurons via enhanced endocytosis of tau[19]. Thus, alterations in BIN1 may promote prion-like spreading of tau within the brain. Alternatively, alterations in BIN1 may be associated with beta-amyloid (Aβ$_{1–42}$)[20]. Post-mortem analyses in AD have shown that BIN1 accumulates in the vicinity of amyloid plaques[21]. In primary neuronal cultures, BIN1 was found to regulate the intraneuronal cleavage of the amyloid precursor protein (APP) by β-secretase[22]. However, the BIN1-mediated increase in the intraneuronal pool of Aβ$_{1–42}$ only weakly translated into higher extraneuronal levels of Aβ$_{1–42}$, i.e., a core feature of AD, and several histochemical brain autopsy and cell culture studies suggested that alterations in BIN1 expression were associated with stronger tau pathology rather than Aβ[15,16]. Together, these findings from preclinical studies suggest that the risk conferred by BIN1 genetic variants may be exerted via promoting tau pathology rather than Aβ in the brain.

Up to now, the translation of these findings to patients with AD using in vivo biomarkers of tau pathology has been difficult. In mild cognitive impairment (MCI) and AD dementia, the BIN1 rs744373 SNP was reported to be associated with higher cerebrospinal fluid (CSF) levels of total and phospho-tau but not with Aβ[23], whereas others could not detect an association between the BIN1 rs744373 SNP and CSF tau or phospho-tau levels[24]. However, CSF-phospho-tau levels are only moderately associated with neurofibrillary tangles in the brain as assessed post-mortem[25] or by tau-PET imaging[26,27]. Furthermore, CSF tau levels may reflect differences in tau production rather than the amount of pathological tau deposits in the brain[28]. A previous MRI neuroimaging study showed that a BIN1 SNP was associated with decreased cortical thickness in the entorhinal cortex and temporal pole, i.e., sites of early tau pathology[29]. However, that study did not assess tau pathology itself. Thus, the question remains, whether the BIN1 rs744373 SNP is associated with increased tau pathology in subjects with AD.

The introduction of AV1451 tau-PET imaging allows to assess fibrillary tangles in the living brain[30]. Here we employ AV1451 PET imaging in elderly subjects in order to translate previous preclinical and post-mortem findings on the association BIN1 and primary AD pathology. We assess whether carriers of the BIN1 rs744373 SNP show elevated regional levels of AV1451 tau-PET in those brain regions that are known to show increased susceptibility to tau pathology as defined by the post-mortem established Braak staging[31]. We test the associations of the BIN1 rs744373 SNP in non-demented subjects to understand whether the BIN1 SNP is associated with tau in the early stages of the development of tau pathology. In addition, given previous evidence of a potential involvement of BIN1 rs744373 in Aβ pathology[21,23], we assess whether the BIN1 rs744373 SNP is associated with higher regional Aβ deposition as assessed by AV45-PET in the same subjects. We hypothesize that carriage of the BIN1 rs744373 risk-allele selectively enhances tau pathology. Since BIN1 genetic variants were previously associated with faster cognitive decline, we lastly test whether alterations in AV1451 tau-PET levels mediate the association between the BIN1 rs744373 SNP and worse memory performance.

## Results

**Sample characteristics**. For the current study, we analyzed data from 89 participants of the ADNI cohort, including 49 cognitively normal (CN) and 40 mild cognitively impaired (MCI) subjects (see Table 1 for sample characteristics). All subjects underwent AV1451 tau-PET, AV45-amyloid PET, structural MRI and cognitive testing at the same study visit of ADNI phase 3. The genotype of the BIN1 rs744373 SNP was extracted from ADNI GWAS data provided by the ADNI genetics core, where we found 22 CN subjects and 18 MCI subjects to carry at least one copy of the BIN1 rs744373 G-allele which confers higher risk of AD dementia as shown by GWAS[4]. Henceforth, these subjects will be referred to as BIN1 rs744373 risk-allele carriers. BIN1 rs744373 allele distribution (GG/GA/AA = 8/32/49) did not deviate from Hardy–Weinberg equilibrium ($p = 0.422$, Chi-squared test). There were no differences in baseline demographics (age, gender, education) between CN vs. MCI or between BIN1 rs744373 risk-allele vs. normal-allele carriers. In total, 48 subjects (24 CN &, 24 MCI) showed abnormally elevated Aβ-levels as determined via AV45 PET (i.e., global standardized uptake value ratio (SUVR) > 1.11).

***BIN1* rs744373 is associated with higher tau-PET uptake**. In a first step, we tested the hypothesis that BIN1 rs744373 risk-allele carriers show higher tau pathology (i.e., global or for regions corresponding to Braak stages I–VI, Fig. 1a) than carriers of the normal BIN1 rs744373 allele. For global tau, we quantified the overall tau load as global AV1451 tau-PET uptake using an established Freesurfer-based protocol (i.e., standardized volume uptake ratio normalized to the inferior cerebellar gray)[32]. When testing via ANCOVA whether the BIN1 rs744373 SNP had an effect on global AV1451 tau-PET uptake, we found risk-allele carriers to show elevated global tau levels with a Cohens d of 0.562, controlling for age, gender, education, ApoE ε4 status, diagnosis, and gray matter (GM) volume of the global tau ROI

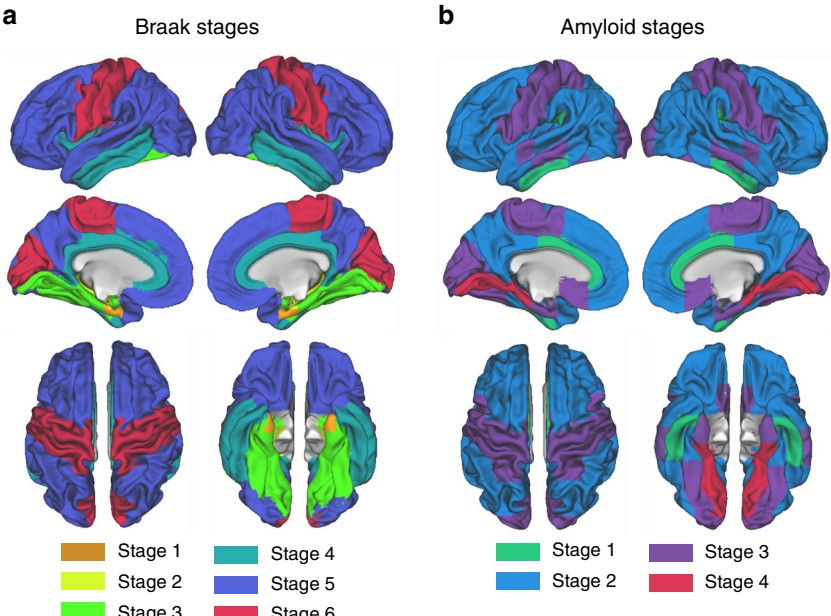

**Fig. 1** Staging systems for tau- and amyloid-PET. Spatial mapping of Braak- (**a**) and amyloid stage-specific ROIs (**b**) that were used to determine regional AV1451 tau- and AV45 amyloid-PET uptake within the sample of $n = 89$ subjects

($F_{(81,7)} = 7.694$, $p = 0.007$, see Fig. 2a). To test whether the effects of BIN1 rs744373 on AV1451 tau-PET uptake were independent of amyloid pathology, we further included overall amyloid load as a covariate, which was quantified as the global AV45 amyloid-PET uptake. Importantly, inclusion of global AV45 amyloid-PET uptake as a covariate did not alter the association between BIN1 rs744373 risk-allele carriage and elevated global AV1451 tau-PET uptake ($F_{(80,8)} = 7.658$, $p = 0.007$, ANCOVA, see model 2 in Table 2 for statistics). Results remained unchanged when alternatively including Aβ-status as a covariate. This suggests that the associations between BIN1 rs744373 and increased global AV1451 tau-PET uptake are independent of diagnostic group or Aβ-status. Results for the main analysis (i.e., SNP effect on global AV1451 tau-PET) were consistent when repeated for the previously reported BIN1 rs7561528 risk variant[14], which was also available from ADNI GWAS data (48 risk-allele carriers vs. 41 normal-allele carriers, $F_{(80,8)} = 3.217$, $p = 0.045$, ANCOVA)[14].

We next tested, whether the effects of BIN1 rs744373 on tau pathology showed regional differences. To this end, we assessed the AV1451 tau-PET SUVR within brain regions corresponding to Braak stages I–VI (Fig. 1a) that recapitulate the spatial tau-spreading pattern from early-to-late-stage tau pathology across the cortex[33]. Here, we could consistently detect significantly ($p < 0.05$, ANCOVA) elevated tau load in BIN1 rs744373 risk-allele carriers across regions corresponding to Braak stages 2–6, with effect sizes ranging between 0.430 and 0.594. This suggests that the BIN1 rs744373 risk allele is associated with general brain-wide increases in tau. These analyses are summarized in Fig. 2a and Table 2. Despite the highly consistent effects of the BIN1 rs744373 SNP on tau, only the associations for global and Braak stage 5 AV1451 tau-PET SUVR remained significant after applying a Bonferroni-corrected α-threshold of 0.0071 (i.e., α = 0.05 adjusted for 7 tests). Again, these results remained fully consistent when additionally controlling for amyloid levels (i.e., global AV45 amyloid-PET SUVR) as summarized in Table 2 (shown as model 2). Results remained also unchanged when controlling for Aβ status. These findings suggest that the effects of

BIN1 rs744373 on AV1451 tau-PET uptake are independent of Aβ or diagnosis.

When testing, whether the BIN1 rs744373 SNP had an effect on amyloid load, we did not detect any differences between risk-allele and normal-allele carriers for global AV45 amyloid-PET SUVR, controlling for age, gender, education, ApoE ε4 status, diagnosis, and GM volume of the respective amyloid ROI ($F_{(81,7)} = 0.148$, $p = 0.701$, ANCOVA). To address potential effects of BIN1 rs744373 on regional amyloid levels, we quantified amyloid SUVRs within four distinct early- to late-amyloid stage ROIs (Fig. 1b) that have been previously shown to recapitulate amyloid spread[34]. In line with the results for global AV45 amyloid-PET uptake, we found no significant effect of the BIN1 rs744373 SNP on AV45 amyloid-PET uptake within the different amyloid stage ROIs (all $p > 0.05$, ANCOVAs, see Fig. 2b & Table 3 for statistics), when controlling for age, gender, education, ApoE ε4 status, diagnosis, and GM volume of the, respective, amyloid-stage ROIs. All results remained consistent when additionally including global AV1451 tau-PET SUVRs as a covariate (see Table 3). Again, there was no interaction between diagnosis and BIN1 rs744373 or between global AV45 tau-PET uptake and BIN1 rs744373 on AV45 amyloid-PET uptake.

**BIN1 rs744373 effects on tau are independent of amyloid.** Next, we tested whether BIN1 rs744373 risk-allele carriage was associated with higher tau independent of the level of amyloid. Using linear regression controlling for age, gender diagnosis and ApoE ε4 status, we found that both global AV45 amyloid-PET SUVR ($t_{(81)} = 2.506$, $β = 0.248$, $p = 0.014$) and BIN1 rs744373 ($t_{(81)} = 2.882$, $β = 0.280$, $p = 0.005$) had independent main effects on global AV1451 tau-PET SUVR. No interaction between Aβ status (binary) or global AV45 amyloid-PET SUVR (continuous) and BIN1 rs744373 on AV1451 tau-PET was found. To illustrate the main effect of BIN1 rs744373 effects on AV1451 tau-PET that was independent of AV45 amyloid-PET, we have plotted the association between BIN1 rs744373 and global AV1451 tau-PET uptake at each quartile of global AV45 amyloid-PET (Fig. 3).

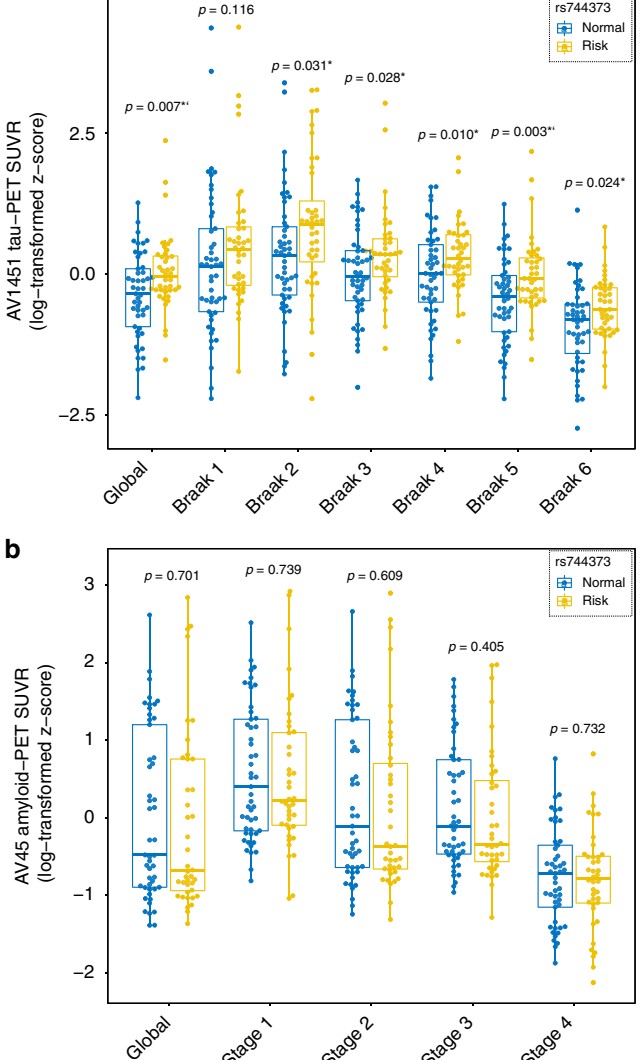

**Fig. 2** Effects of BIN1 rs744373 on tau- and amyloid-PET uptake. **a** Boxplots showing the differences in global or regional AV1451 tau-PET SUVRs between BIN1 rs744373 risk allele ($n = 40$, yellow) vs. normal-allele carriers ($n = 49$, blue). P-values are based on ANCOVA models controlled for age, gender, education, diagnosis, memory performance (i.e., ADNI-MEM) and gray matter volume of the respective ROI. **b** Differences in global or regional AV45 amyloid-PET uptake between BIN1 rs744373 risk allele ($n = 40$, yellow) vs. normal-allele carriers ($n = 49$, blue). Boxplots are displayed as median (center line) ±interquartile range (box boundaries) with whiskers including observations falling within the 1.5 interquartile range. P-values are again derived from ANCOVA models controlled for age, gender, education, diagnosis, ApoE ε4 carrier status, and gray matter volume of the respective ROI. *$p < 0.05$ (uncorrected); ' = significant after Bonferroni correction ($p < 0.0071$)

**Spatial match of _BIN1_ mRNA expression and tau pathology**. Our finding of the association between _BIN1_ rs744373 and higher AV1451 tau-PET levels, plus previous reports of altered _BIN1_ mRNA levels in AD[15] suggest that alterations in cerebral _BIN1_ expression are related to tau pathology. In order to further substantiate this hypothesis, we argued that regional expression patterns of _BIN1_ in the brain should overlap with those brain regions of increased vulnerability to tau pathology, given the previously reported association between _BIN1_ mRNA expression and neurofibrillary tangles in post-mortem brains[15,16]. In order to

test this hypothesis, we obtained whole-brain mRNA expression levels of _BIN1_ from the Allen Brain Atlas, which is based on post-mortem brain-wide microarray assessments of six healthy brain donors without any history of psychiatric or neurological disorders[35,36]. Specifically, we used the median of microarray-based $\log_2$ mRNA expression of _BIN1_ across the six donors that have been recently summarized for each region of the Freesurfer-based Desikan–Killiany atlas[37]. For the same atlas regions, we determined the group-median AV1451 tau-PET SUVR across all subjects in the current study. We restricted the analysis to the left hemisphere since full mRNA expression data of all six brain donors is only available for this hemisphere. Surface renderings of both group average left hemispheric tau and _BIN1_ mRNA expression are depicted in the upper panel of Fig. 4. We used spatial regression on 10,000 bootstrapped samples based on which group-median tau levels were iteratively determined to assess the association between group-median AV1451 tau-PET SUVR and _BIN1_ mRNA expression patterns. Here, we found a significant average positive Pearson-Moment correlation ($r = 0.374$, see Fig. 5) with a 95% CI of [0.370:0.379] ($p < 0.001$). These results suggest that regions with higher _BIN1_ mRNA expression have a higher likelihood of showing elevated tau levels. In order to assess the spatial overlap of regions with high _BIN1_ expression or high AV1451 tau-PET uptake, we thresholded both the group-median tau PET and the BIN1 mRNA expression maps at a percentile threshold of 75%, (Fig. 4 lower panel). Visual inspection suggests that mesio- and inferior temporal brain regions corresponding to early Braak stages show both high _BIN1_ mRNA expression as well as high AV1451 tau-PET uptake.

**Tau mediates _BIN1_ rs744373 effects on memory impairment**. To assess whether the _BIN1_ rs744373 SNP is detrimental for cognition via increasing tau pathology, we tested whether risk-allele carriage is associated with worse memory and whether these effects are mediated via increased tau pathology. To this end, we applied causal mediation analysis with 10 000 bootstrapping iterations controlling for age, gender, education, diagnosis, global AV45 amyloid-PET SUVR, and ApoE ε4 status. Memory performance was assessed based on ADNI-MEM, which is an established composite score developed by the ADNI core that summarizes the performance on multiple memory tests[38]. Supporting our hypothesis, we found that the _BIN1_ rs744373 risk-allele was significantly associated with worse ADNI-MEM scores ($\beta = -0.25$, $p = 0.030$), where this association was mediated via global AV1451 tau-PET uptake (bootstrapped average causal mediation effect: $\beta = -0.083$ [$-0.180;0.00$], $p = 0.016$). The effect was considered a full mediation, since the direct effect of the _BIN1_ rs744373 risk allele on ADNI-MEM was no longer significant ($\beta = -0.17$, $p = 0.15$) in the presence of the mediator (i.e., global AV1451 tau-PET uptake). A path model of this mediation analysis is shown in Fig. 6.

**Discussion**
The major finding of the current study was that the _BIN1_ rs744373 risk allele was associated elevated AV1451 tau-PET uptake. In contrast, we detected no association between _BIN1_ rs744373 and regional Aβ assessed by AV45-PET, suggesting that _BIN1_ is associated with higher tau pathology rather than Aβ. In addition, we found _BIN1_ rs744373 to be associated with worse memory performance, where this effect was mediated by _BIN1_ rs744373-associated elevation of global AV1451 tau-PET. Together, our findings support the hypothesis that the _BIN1_ rs744373 risk-allele is associated with elevated cerebral tau pathology, thereby worsening memory decline. Our findings represent an important contribution to the understanding of the role of _BIN1_

**Table 1 Sample characteristics**

|  | CN BIN1 Normal ($n = 27$) | CN BIN1 Risk ($n = 22$) | MCI-BIN1 normal ($n = 22$) | MCI-BIN1 risk ($n = 18$) | p-value |
|---|---|---|---|---|---|
| Age | 80.3 (6.09) | 80.25 (5.55) | 76.48 (8.13) | 77.03 (6.35) | 0.118 |
| Gender (m/f) | 14/13 | 11/11 | 11/11 | 10/8 | 0.984 |
| Education | 17.07 (2.38) | 16.64 (2.94) | 15.45 (3.47) | 15.78 (2.67) | 0.349 |
| Aβ-status (pos/neg) | 12/15 | 12/10 | 12/10 | 12/6 | 0.846 |
| ApoE ε4 pos/neg | 8/19 | 5/17 | 8/14 | 3/15 | 0.587 |
| MMSE | 29.15 (1.19) | 28.41 (2.22) | 27.95 (2.06) | 28.22 (1.40) | 0.112 |
| ADNI-MEM | 0.93 (0.51)[c,d] | 0.79 (0.43)[d] | 0.43 (0.75)[a] | 0.09 (0.43)[a,b] | <0.001 |
| AV45 global SUVR | 1.15 (0.23) | 1.15 (0.24) | 1.18 (0.21) | 1.13 (0.26) | 0.929 |
| AV1451 global SUVR | 1.05 (0.12)[d] | 1.09 (0.07)[d] | 1.08 (0.10)[d] | 1.18 (0.15)[a,b,c] | 0.002 |

*CN* Cognitively Normal, *MCI* Mild Cognitive Impairment, *M* male, *f* female, *MMSE* Mini-Mental State Exam, *ADNI-MEM* Alzheimer's Disease Neuroimaging Initiative-Memory composite
[a] sig. (p < 0.05) different from CN-BIN1 normal
[b] sig. different from CN-BIN1 risk
[c] sig. different from MCI-BIN1 normal
[d] sig. different from MCI-BIN1 risk

**Table 2 BIN1 rs744373 risk allele as a predictor of AV1451 tau-PET SUVR**

| Dependent variable | AV1451 SUVR: rs744373 risk | AV1451 SUVR: rs744373 normal | Model 1: F | Model 1: P | Model 2: F | Model 2: p | Cohens d |
|---|---|---|---|---|---|---|---|
| Global | 1.13 (0.12) | 1.06 (0.12) | 7.694 | 0.007[*a] | 7.658 | 0.007[*a] | 0.562 |
| Braak 1 | 1.23 (0.23) | 1.17 (0.24) | 2.526 | 0.116 | 2. 496 | 0.118 | 0.250 |
| Braak 2 | 1.28 (0.23) | 1.18 (0.20) | 4.809 | 0.031[*] | 4.749 | 0.032[*] | 0.465 |
| Braak 3 | 1.18 (0.15) | 1.13 (0.13) | 4.996 | 0.028[*] | 4.992 | 0.028[*] | 0.430 |
| Braak 4 | 1.19 (0.16) | 1.12 (0.13) | 6.920 | 0.010[*] | 6.883 | 0.010[*] | 0.496 |
| Braak 5 | 1.12 (0.12) | 1.05 (0.11) | 9.155 | 0.003[a] | 9.087 | 0.003[*a] | 0.594 |
| Braak 6 | 1.03 (0.08) | 0.98 (0.11) | 5.330 | 0.024[*] | 5.263 | 0.024[*] | 0.463 |

Model 1 Covariates: age, gender, education, diagnosis, ROI gray matter, ApoE ε4. Model 2 Covariates: Global AV45 amyloid-PET, age, gender, education, diagnosis, ROI gray matter, ApoE ε4
[*]Significant at p < 0.05 (uncorrected)
[a]Significant after Bonferroni correction for 7 tests (p < 0.0071)

**Table 3 BIN1 rs744373 risk allele as a predictor of AV45 amyloid-PET SUVR**

| Dependent variable | AV45 SUVR: rs744373 risk | AV45 SUVR: rs744373 normal | Model 1: F | Model 1: P | Model 2: F | Model 2: P | Cohens d |
|---|---|---|---|---|---|---|---|
| Global | 1.14 (0.24) | 1.16 (0.22) | 0.148 | 0.701 | 0.148 | 0.702 | 0.079 |
| Stage 1 | 1.25 (0.21) | 1.26 (0.19) | 0.111 | 0.739 | 0.111 | 0.740 | 0.052 |
| Stage 2 | 1.18 (0.23) | 1.20 (0.21) | 0.263 | 0.609 | 0.261 | 0.610 | 0.086 |
| Stage 3 | 1.15 (0.17) | 1.17 (0.16) | 0.701 | 0.405 | 0.694 | 0.407 | 0.156 |
| Stage 4 | 1.00 (0.11) | 1.01 (0.10) | 0.118 | 0.732 | 0.118 | 0.732 | 0.068 |

Model 1 Covariates: age, gender, education, diagnosis, ROI gray matter, ApoE ε4. Model 2 Covariates: Global AV1451 tau-PET, age, gender, education, diagnosis, ROI gray matter, ApoE ε4

in AD, as we demonstrate in living non-demented elderly subjects an association of *BIN1* rs744373 and regional elevation of tau pathology, i.e., a key AD pathology associated with cognitive impairment.

For our first finding, the association between *BIN1* rs744373 and higher AV1451 tau-PET ROI values but not AV45 amyloid-PET, suggests a selective association between the *BIN1* rs744373 SNP and PET-assessed tau pathology. Of note, we did not find an interaction between *BIN1* rs744373 and AV45 amyloid-PET on AV1451 tau-PET levels, suggesting that the association between *BIN1* and tau does not depend on the presence of Aβ pathology. Supporting this notion, effects of *BIN1* rs744373 on AV1451 tau-PET levels remained consistent when controlled for Aβ-status or continuous AV45 amyloid-PET levels. These findings are in general agreement with several previous findings suggesting that BIN1 is linked to tau pathology rather than amyloid pathology. Here, it has been previously reported that brain BIN1 protein levels are correlated

with neurofibrillary tangle pathology but not with diffuse or neuritic amyloid plaques in AD brains[16]. In a similar vein, *BIN1* risk variants have been previously shown to correlate with post-mortem assessed brain levels of AT8 positive tau pathology but not with Aβ[15]. In contrast, one previous study showed that BIN1 becomes insoluble and accumulates in the vicinity of amyloid plaques in a mouse model of AD and in brain sections from AD patients[39]. Still, it is unclear whether BIN1 alterations are a cause or a consequence of amyloid pathology. Our current results suggest that the *BIN1* rs744373 risk allele is associated primarily with PET-assessed tau rather than amyloid levels.

The current results are also in agreement with previous reports of *BIN1* rs744373 risk-allele carriage being associated with increased CSF-phospho-tau and CSF-total tau levels but not with CSF-Aβ levels and amyloid PET[23]. We caution however that our current finding of absence of an association between *BIN1* rs744373 and AV45 amyloid-PET may partially be due to the

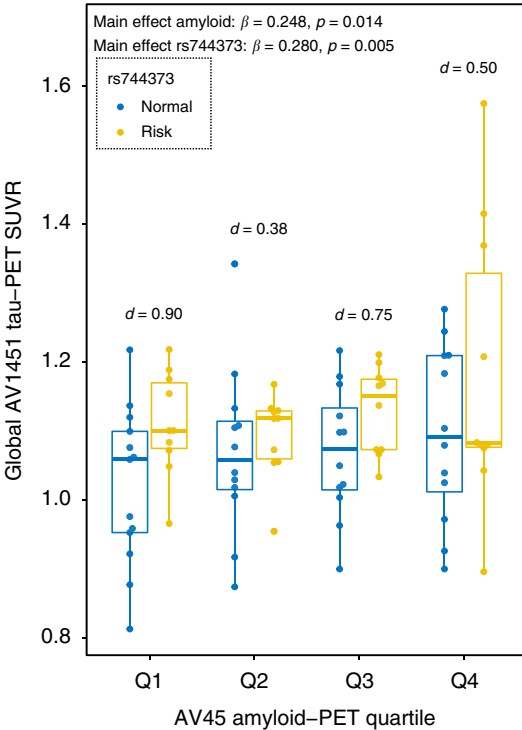

**Fig. 3** BIN1 rs744373 effects on tau across the amyloid spectrum. Boxplot showing the association between BIN1 rs744373 (risk-allele $n = 40$, yellow; normal-allele $n = 49$, blue) and global AV1451 tau-PET uptake across amyloid quartiles. The statistical main effects presented in the upper left corner of the graph are derived from linear regression, with the BIN1 rs7443733 SNP and global AV45 amyloid-PET SUVR as predictors, controlling for age, gender, diagnosis, and ApoE ε4 status. Cohens d effects sizes displayed in the plot were derived for each quartile. Note, that no interaction between BIN1 rs744373 and global AV45 amyloid-PET uptake was found, suggesting that the effects of BIN1 rs744373 on tau are similar between high- and low-amyloid groups. Boxplots are displayed as median (center line) ±interquartile range (box boundaries) with whiskers including observations falling within the 1.5 interquartile range

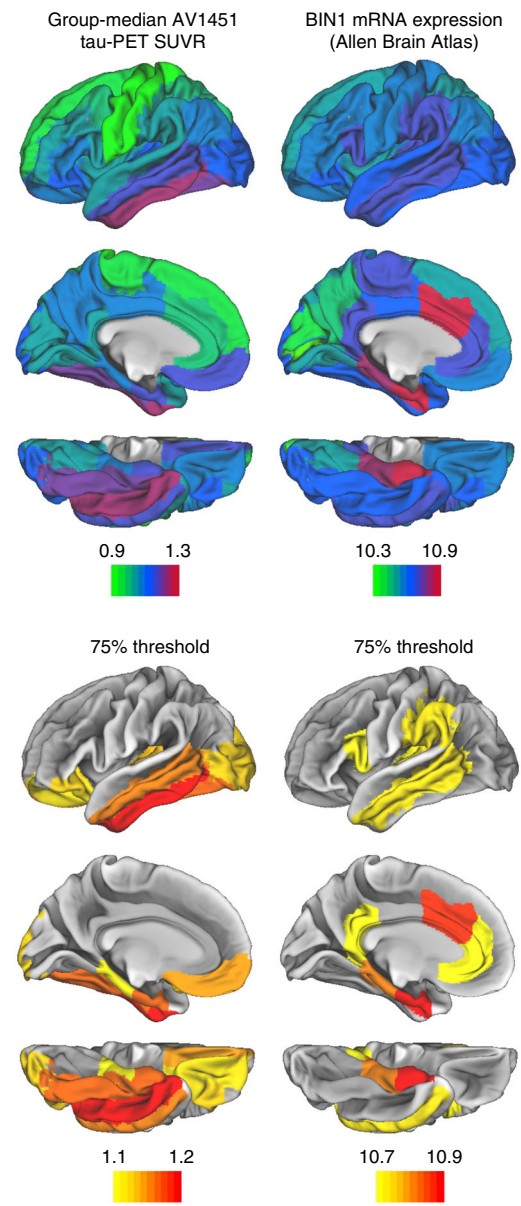

**Fig. 4** Spatial patterns of tau-PET and BIN1 mRNA expression. Spatial mapping of median BIN1 mRNA expression (i.e., $\log_2$) derived from the Allen Brain Atlas and group-median AV1451 tau-PET SUVR (derived from $n = 89$ subjects), either for all ROIs (upper panel) or restricted to regions falling above the 75th percentile of either BIN1 mRNA expression or group-median AV1451 tau-PET uptake (lower panel). Color scales represent SUVR scores for AV1451 tau-PET and $\log_2$ mRNA expression for BIN1 mRNA

fact that in many subjects amyloid may have reached already plateau levels[40], leading to reduced variability in AV45 amyloid-PET levels. Thus, it is still possible that BIN1 is associated with Aβ pathology especially at the early presymptomatic phase before a plateau is reached.

The molecular mechanisms underlying the association between *BIN1* genetic variants and increased AD risk are not known. For the link between BIN1 and tau pathology, in vitro studies showed that BIN1 binds to tau via a proline rich SH3 domain[15,41–43], possibly reducing the integrity of the cytoskeleton[41,42]. Our findings of a spatial match between the *BIN1* mRNA expression pattern in the brain and regions of increased AV1451 tau-PET uptake support the notion that *BIN1* SNP-related alterations of BIN1 expression are associated with tau pathology. It is unclear, however, whether the binding of BIN1 to tau entails the development of pathological fibrillary tau[15]. Alternatively, BIN1 may enhance spreading of pathological tau across connected neurons[44] via endocytosis, i.e., a pathway that has recently been suggested to lead to prion-like spreading of tau in the brain[45,46]. The neuron-specific BIN1 isoform 1 interacts with clathrin, thereby attenuating the post-synaptic endocytotic uptake of tau[19,42]. Recent histological brain-autopsy studies showed that while BIN1 protein expression is overall increased in AD, the BIN1 isoform 1 is decreased in AD[16]. Together, these results

suggest that reduced BIN1 isoform 1 levels may enhance the endocytosis-mediated spreading of tau pathology in AD. However, it is unknown whether the *BIN1* genetic variants are associated with reduced BIN1 isoform 1 expression and future studies need to clarify the exact pathomechanisms of BIN1 alterations.

Our second major finding was that the association between *BIN1* rs744373 and memory impairment was mediated via elevated global tau levels. This suggests that *BIN1* rs744373 contributes to the development of tau pathology in at-risk subjects, resulting in stronger cognitive impairment. These results are in agreement with previous findings of *BIN1* rs744373 being associated with faster decline in global cognition[47] and episodic memory[48]. Our results are also consistent with previous studies

showing a close association between tau PET and cognitive decline[33,49]. Our mediation analysis suggests that the *BIN1* rs744373 SNP is linked to memory impairment due to the increase in tau pathology in the *BIN1* rs744373 risk-allele carriers. We caution that the study design is correlational in nature and thus a causative interpretation should not be drawn. However, our findings provide a putative pathomechanistic link between the *BIN1* rs744373 SNP and the increase in dementia risk as reported by GWAS[3].

We point out several caveats that should be considered when interpreting the results of the current study. Firstly, there are several *BIN1* genetic variants associated with an increased risk of AD[9,14,15].

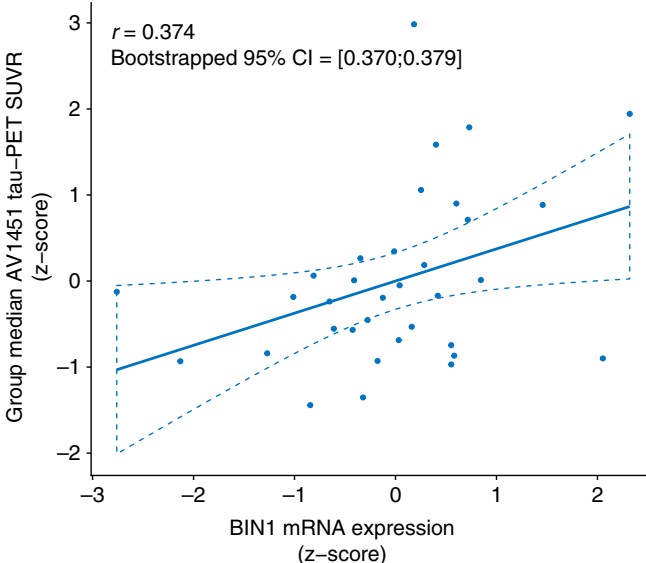

**Fig. 5** Association between tau-PET and BIN1 mRNA expression. Scatterplot showing the positive association between ROI-based BIN1 mRNA expression obtained from the Allen Brain Atlas and group-median AV1451 tau-PET uptake (Number of ROIs = 34; Association illustrated as least-squares regression line ±95% confidence interval). *P*-values and 95% confidence intervals of *r*-values are derived from bootstrapped spatial regression with 1000 iterations

The different SNPs on the *BIN1* gene may be in disequilibrium and may add independently from each other to the risk of AD[14]. Here, we focused on rs744373 because this SNP is most frequently reported to be associated with AD across different GWAS studies[3,7] (see also AlzGene database at http://www.alzgene.org/)[50]. Although rs744373 is the primary *BIN1* SNP associated with increased AD risk, a previous study suggested that the Indel rs59335482 is associated with increased *BIN1* mRNA in post-mortem analyzed brains from AD patients, suggesting that rs59335482 is the functionally effective *BIN1* genetic variant associated with AD risk. However, the Indel rs59335482, which was not available in the current GWAS, is in almost complete linkage equilibrium with rs744373[15], suggesting both SNPs share redundant predictive value. Furthermore, control analysis using an alternative SNP rs7561528 that was available from the GWAS analysis, i.e., another frequently reported *BIN1* SNP as a risk factor of AD[14], confirmed the association between the *BIN1* SNP and increased regional AV1451 tau-PET, suggesting that the current findings were not specific to rs744373 as a tau-related genetic variant of *BIN1*.

Secondly, we caution that even though the current findings suggests that *BIN1* rs744373 is associated with tau pathology, other pathomechanisms of BIN1 may contribute to the increased risk of AD[51]. The BIN1 protein is highly expressed in the white matter (WM) and oligodendrocytes[52] and BIN1 is a key driver of a oligodendrocyte-associated genetic co-expression network that is dysregulated in AD[53]. These results suggest that BIN1 is associated with altered oligodendrocyte integrity that may contribute to white-matter alterations that are a core part of AD-related pathological changes[54,55]. Also, BIN1 has been shown to interact with physiological tau in vitro, i.e., a key constituent of microtubules[15]. It is thus possible, that alterations of BIN1 in the white matter increase the development of pathological tau, which manifests distantly as neurofibrillary tau tangles in the soma. The current results are not in conflict with those alternative pathomechanisms, as the ubiquitously expressed BIN1 protein subserves multiple diverse functions in the brain[12,14], and may thus be involved in multiple AD-related pathological pathways.

Thirdly, the AV1451 tau-PET tracer has previously shown off-target binding in the meninges, basal ganglia, and choroid plexus, which may confound the assessment of tau pathology in cortical and subcortical brain regions[56]. To address this, we excluded ROIs covering the basal ganglia in our analysis to avoid known off-target

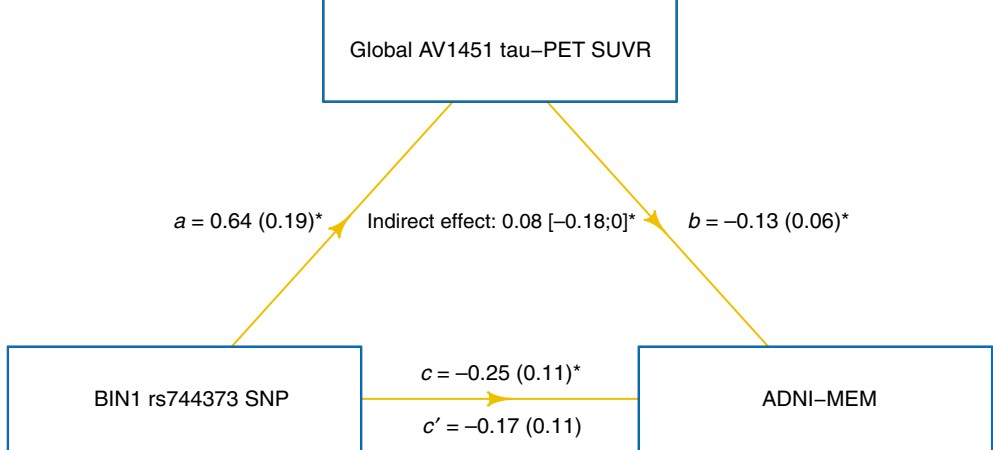

**Fig. 6** Tau mediates effects of BIN1 rs744373 on memory impairment. Path diagram of the mediation model (assessed on the full sample of *n* = 89 subjects), showing that associations between BIN1 rs744373 risk-allele carriage and worse memory are mediated via global AV1451 tau-PET uptake. Path-weights are displayed as beta values with standard errors in brackets. All paths are controlled for age, gender, education, diagnosis, global AV45 amyloid-PET uptake, and ApoE ε4 carrier status. Asterisks indicate *p*-values below 0.05. Significance of the indirect effect was determined using bootstrapping with 10,000 iterations

binding sites to confound our analyses. However, in the current study the AV1451 signal within the hippocampus may be affected by off-target binding in the choroid plexus, hence these results await replication using second generation tau tracers with a better off-target binding profile. Importantly, however, the *BIN1* rs744373 association with higher tau was not limited to the hippocampal AV1451 tau-PET, but was widespread within the brain, rendering it highly unlikely that spill-off from regions of unspecific AV1451 tracer binding accounted for the current results.

In conclusion, our results show an association between *BIN1* rs744373 risk-allele carriage and in vivo assessed tau pathology in elderly subjects with and without amyloid pathology. The current findings suggest that tau pathology provides a key link that underlies the association between *BIN1* genetic variants and cognitive impairment. Future studies may further test whether *BIN1* risk variants are associated with higher rates of increases in pathological tau and faster cognitive decline and conversion to dementia. It is currently unclear whether BIN1 genetic variants are associated with tauopathies other than AD. A recent study reported no association between BIN rs744373 and fronto-temporal dementia[57], however, the association between BIN1 and tau pathology in conditions other than AD remains to be tested in the future. From a clinical point of view, our findings encourage future research in targeting BIN1 protein modification as a potential therapeutic approach to reduce levels of tau pathology. Preclinical studies showed that knock-down of BIN1 reduced tau-related toxicity in a drosophila model of neurodegeneration[15]. Understanding the exact mechanisms of the association between BIN1 and tau pathology are of pivotal clinical importance, since tau is the best predictor of clinical severity in contrast to Aβ deposition[58]. Neuroimaging of tau PET may be a marker to monitor treatment effects of disease modifying therapies targeting BIN1.

## Methods

**Study design**. We included 89 participants from ADNI phase 3 (ClinicalTrials.gov ID: NCT02854033) in whom [18]F-AV1451 tau-PET was obtained. [18]F-AV1451 PET was added only in phase 3 of ADNI, and is thus only available in a smaller subset of the large ADNI cohort. The current set of 89 subjects resulted from inclusion criteria of the availability of T1-weighted MRI, [18]F-AV45 amyloid-PET, cognitive and GWAS data in addition to [18]F-AV1451 tau-PET. All imaging modalities had to be obtained at the same study visit. Selection bias was tested against the entire ADNI cohort of 1784 subjects. Here, we found no differences in gender or education between our selected sample and the entire ADNI sample, however, the mean age of the selected sample (~78.7 years) was significantly ($p < 0.05$) higher than the mean age of the entire ADNI cohort (~73.7 years). Subjects were clinically classified by ADNI centers as cognitively normal (CN, MMSE > 24, CDR = 0, non-depressed) or mild cognitively impaired (MCI; MMSE > 24, CDR = 0.5, objective memory-loss on the education adjusted Wechsler Memory Scale II, preserved activities of daily living)[59]. Subjects with AD dementia were excluded from the analysis due to the small number of cases that met our inclusion criteria ($n = 3$). The *BIN1* rs744373 genotype was extracted from GWAS data provided by the ADNI genetics core, where whole-genome sequencing was conducted using the Ilumina Omni 2.5 M Bead Chip. For a detailed description of the whole-genome sequencing methods, please refer to a previous publication by the ADNI genetics core[60]. Subjects were assigned to the *BIN1* rs744373 risk-group ($n = 40$) when carrying at least one G-allele[14]. Ethical approval was obtained by the ADNI investigators at each study site, all participants provided written informed consent and all work complied with ethical regulations for work with human participants.

**Image acquisition**. All imaging data was downloaded from the ADNI loni image archive (https://ida.loni.usc.edu). Structural MRI in ADNI3 was recorded using a 3D T1-weighted MPRAGE sequence with 1 mm isotropic voxel-space and a TR = 2300 ms. Detailed sequence parameters can be found online at (http://adni.loni.usc.edu/wp-content/uploads/2017/07/ADNI3-MRI-protocols.pdf).

[18]F-AV1451 tau-PET was acquired 75–105 min post-injection of [18]F-AV1451, in 6 × 5 minute time frames. [18]F-AV-45 Florbetapir amyloid PET scans were obtained during 4 × 5 min time frames measured 50–70 min post-injection of [18]F-AV45 (http://adni.loni.usc.edu/wp-content/uploads/2010/05/ADNI2_PET_Tech_Manual_0142011.pdf).

For both AV1451 tau- and AV45 amyloid-PET we downloaded partially preprocessed data where dynamically acquired image frames are first registered to an AC-PC orientation and standard voxel image grid and subsequently averaged to obtain a single image for each PET modality. Having PET images in a standard image matrix facilitates the combination of PET images from different scanners. For further details please see refer to the ADNI website (http://adni.loni.usc.edu/methods/pet-analysis-method/pet-analysis/).

**Image preprocessing**. All MRI and PET images were inspected for artifacts prior to preprocessing. We applied two processing pipelines to PET data, following pre-established protocols to evaluate regional and global AV1451 tau- and AV45 amyloid-PET uptake. First, we used a SPM12-based pipeline to obtain stage-specific AV45 amyloid-PET SUVR scores[34]. Second, we applied a Freesurfer-based pipeline to obtain global AV45 amyloid-[61] as well as global and regional AV1451 tau-PET SUVRs (i.e., Braak stage ROIs)[33].

For the pre-established SPM12-based pipeline[62–67], native-space structural MRI images were first segmented into gray matter (GM), white matter (WM) and CSF maps using SPMs new segment approach. Using SPMs high-dimensional DARTEL warping algorithm, we estimated subject-specific flow-fields to non-linearly transform all GM, WM, and CSF maps to a sample-specific template that was determined in an iterative procedure. Using affine transformation, this sample-specific template was subsequently normalized to Montreal Neurological Institute (MNI) standard space. Next, subject-specific AV45 amyloid-PET images were co-registered to the corresponding high-resolution T1 image and subsequently DARTEL warped to MNI standard space. We did not spatially smooth the images to avoid spill over between adjacent regions during ROI-based analyses.

For the Freesurfer-based pipeline (Version 5.3), we applied volumetric segmentation to the high-resolution native-space structural MRI images, where subcortical and cortical areas are segmented automatically using the probabilistic Desikan–Killiany Atlas[68]. The segmented anatomical ROIs from the high-resolution structural MRI images were then applied to the co-registered AV45 amyloid- and AV1451 tau-PET images to extract ROI-based values. To obtain SUVR scores, all ROI values were normalized to the mean uptake of the whole cerebellum for AV45 data, and to the mean uptake of the inferior cerebellar gray for AV1451 data, following previous recommendations[32,61].

**Amyloid staging**. For AV45 amyloid-PET, we assessed global amyloid-PET levels that are commonly used for subject stratification into Aβ-positive/negative plus a more fine-grained anatomical amyloid staging system that was introduced recently[34]. For global amyloid load we computed global AV45 amyloid-PET SUVRs using an established Freesurfer pipeline. In brief, we averaged Freesurfer-defined SUVR (normalized to the whole cerebellum) scores across lateral and medial frontal, anterior, and posterior cingulate, lateral parietal and lateral temporal regions[61]. Based on these scores, Aβ-positivity was defined as a global AV45 amyloid-PET SUVR > 1.11[69]. Summary statistics on amyloid status can be found in table 1.

We further assessed local amyloid levels using a 4-stage model, that suggests amyloid deposition to initiate in the temporobasal and mediofrontal areas with subsequent affection of the associative neocortex, primary sensorimotor areas and lastly the basal ganglia[34]. To this end, we used the MNI normalized AV45 amyloid-PET images from our SPM12 pipeline where we determined the mean scores within the four amyloid stage ROIs (shown in Fig. 1b) that were built using the MNI-space based Harvard–Oxford brain atlas following a previously described protocol[34]. Again, these mean values were intensity normalized to Freesurfer derived whole-cerebellar AV45 uptake to obtain SUVR scores.

**Tau staging**. For tau, we also obtained global as well as stage-specific AV1451 tau-PET SUVR scores.

For global tau, we averaged the size-weighted Freesurfer-ROI SUVRs across all Desikan–Killiany atlas regions, excluding the cerebellum, thalamus and basal ganglia (i.e., typical regions of AV1451 off-target binding) following a previously described approach[32]. For stage-specific AV1451 tau-PET uptake, we applied a recently described Braak-ROI staging system that allows application of the post-mortem established tau staging system to tau PET imaging[33]. Here, we obtained size-weighted Freesurfer-ROI SUVRs for each Braak stage ROI, from Braak stage I (i.e., entorhinal cortex) to Braak stage VI (i.e., primary sensorimotor & primary visual cortex). A list of ROIs that are included within each Braak stage ROI can be found online (https://adni.bitbucket.io/reference/docs/UCBERKELEYAV1451/UCBERKELEYAV1451_Methods_FINAL.pdf). A surface rendering of the Braak ROIs is shown in Fig. 1a. Note that we excluded the thalamus or basal ganglia ROIs for all AV1451 tau-PET analyses, due to known off-target binding of the AV1451 tracer in these regions.

**mRNA expression levels of *BIN1***. Regional gene expression was obtained from publicly available microarray measurements of regional mRNA expression based on post-mortem data from the Allen Brain Atlas (http://human.brain-map.org/; RRID: SCR007416). The Allen Brain atlas includes more than 60,000 microarray probes collected from 3700 autopsy-based brain tissue samples from a total of 6 subjects aged 24–57 with no known history of neurological or psychiatric conditions[35,36]. Microarray-based log$_2$ expression values of

20,737 genes within each of the 3700 samples were mapped back into MNI standard space by the Allen Brain Institute using stereotactic coordinates of the examined probes. The whole gene expression data has been recently mapped to the Freesurfer-based Desikan–Killiany atlas as median gene expression for probes falling within each of the 68 atlas ROIs[37]. Here, we specifically extracted median expression of *BIN1* mRNA within these Desikan–Killiany ROIs, to test associations between *BIN1* expression and AV1451 tau-PET uptake. Since microarray assessments and thus *BIN1* mRNA expression of all 6 Allen brain atlas subjects were available only for the left hemisphere (vs. 2 subjects for the right hemisphere), we restricted the analysis of *BIN1* mRNA expression data to the more robust estimates of the left hemisphere in line with previous studies[70–72].

**Statistical analysis**. Group demographics and baseline characteristics were compared between groups (i.e., diagnosis & *BIN1* rs744373 status) using ANOVAs for continuous measures and Chi-squared tests for categorical measures. Global and regional AV45 amyloid- and AV1451 tau-PET SUVR scores were log-transformed prior to analysis to approximate a normal distribution.

For our main analysis, we tested whether *BIN1* rs744373 risk-allele carriage was associated with increased AV1451 tau-PET uptake. To this end, we applied ANCOVAs to test whether presence of the *BIN1* rs744373 risk allele had an effect on global or regional (i.e., Braak stage ROIs) AV1451 tau-PET SUVRs, controlling for age, gender, education, diagnosis, and ApoE ε4 carrier status and GM volume of the respective tau ROI. To assess any effects of the *BIN1* rs744373 SNP on amyloid, we tested the same models this time using global or regional (i.e., amyloid-stage ROIs) AV45 amyloid-PET SUVRs as the dependent variable. Lastly, we tested whether *BIN1* rs744373 risk-allele carriage was associated with higher AV1451 tau-PET SUVR independent of amyloid. To this end, we conducted linear regression with global AV1451 tau-PET SUVR as a dependent variable and global AV45 amyloid-PET SUVR and *BIN1* rs744373 status as independent variables controlling for age, gender, education, ApoE ε4 carrier status, and diagnosis. Using the same covariates as described for the previous model, we further tested the interaction between *BIN1* rs744373 and global AV45 amyloid-PET or Aβ status (Aβ− or Aβ+).

Next, we tested whether higher local *BIN1* mRNA expression levels were associated with an increased likelihood of developing abnormal tau. To this end, we determined *BIN1* mRNA expression using the Allen brain atlas data in Desikan–Killiany atlas space and determined the group-median regional tau PET SUVRs for corresponding anatomical regions. We surface-mapped both group-median tau PET SUVRs and *BIN1* mRNA expression and tested the Pearson–Moment correlation between regional *BIN1* mRNA expression and tau load, applying a two-tailed alpha threshold of 0.05. We repeated this analysis based on 1000 randomly drawn bootstrapped samples to determine the 95% CI of the correlation coefficient between *BIN1* mRNA expression and AV1451 tau-PET uptake.

Lastly, we assessed whether *BIN1* rs744373 risk-allele carriage was associated with worse memory performance, and whether this association was mediated by tau pathology. To test this, we conducted mediation analysis, testing whether the association between *BIN1* rs744373 and ADNI-MEM was mediated via global AV1451 tau-PET uptake. Significance of the mediation effect was determined using 10,000 bootstrapped iterations, where each path of the model was controlled for global AV45 amyloid-PET SUVR, age, gender, education, and diagnosis and ApoE ε4 carrier status.

All statistical analyses were conducted with R statistical software. *P*-values were considered significant when meeting a two-tailed alpha threshold of 0.05. When a single hypothesis was tested multiple times, we also report Bonferroni-corrected *p*-values in case of significant uncorrected p-values.

**Reporting summary**. Further information on experimental design is available in the Nature Research Reporting Summary linked to this article.

## Data availability

The data that support the findings of this study were obtained from the Alzheimer's Disease Neuroimaging Initiative (ADNI) and are available from the ADNI database (adni.loni.usc.edu) upon registration and compliance with the data use agreement. A list including the anonymized participant identifiers of the currently used sample and the source file can be downloaded from the ADNI database (http://adni.loni.usc.edu/). The R-script used for the current study can be obtained from the first author upon request.

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

## Acknowledgements

Data used in preparation of this manuscript were obtained from the ADNI database (adni. loni.usc.edu). As such, the investigators within the ADNI study contributed to the design and implementation of ADNI and/or provided data but did not participate in analysis or writing of this paper. The study was funded by grants from the Alzheimer Forschung Initiative (AFI, Grant 15035 to ME) and European Commission (Grant 334259 to ME). ADNI data collection and sharing for this project was funded by the ADNI (National Institutes of Health Grant U01 AG024904) and DOD ADNI (Department of Defense award number W81XWH-12-2-0012). ADNI is funded by the National Institute on Aging, the National Institute of Biomedical Imaging, and Bioengineering, and through contributions from the following: AbbVie, Alzheimer's Association; Alzheimer's Drug Discovery Foundation; Araclon Biotech; BioClinica, Inc.; Biogen; Bristol-Myers Squibb Company; CereSpir, Inc.; Cogstate; Eisai Inc.; Elan Phar- maceuticals, Inc.; Eli Lilly and Company; EuroImmun; F. Hoffmann-La Roche Ltd and its affiliated company Genentech, Inc.; Fujirebio; GE Healthcare; IXICO Ltd.; Janssen Alzheimer Immunotherapy Research & Development, LLC.; Johnson & Johnson Pharmaceutical Research & Development LLC.; Lumosity; Lundbeck; Merck & Co., Inc.; Meso Scale Diagnostics, LLC.; NeuroRx Research; Neurotrack Technologies; Novartis Pharmaceuticals Corporation; Pfizer Inc.; Piramal Imaging; Servier; Takeda Pharmaceutical Company; and Transition Therapeutics. The Canadian Institutes of Health Research is providing funds to support ADNI clinical sites in Canada. Private sector contributions are facilitated by the Foundation for the National Institutes of Health (www.fnih.org).

## Author contributions

N.F.: study concept and design, data processing, statistical analysis, interpretation of the results, and writing the manuscript. A.R.: data processing and critical revision of the manuscript. J.N.: data processing and critical revision of the manuscript. M.E.: study concept and design, interpretation of the results, and writing the manuscript. ADNI provided all data used for this study.

**Additional information**

**Competing interests:** The authors declare no competing interests.

# The Alzheimer's Disease Neuroimaging Initiative (ADNI)

Michael W. Weiner[2], Paul Aisen[3], Ronald Petersen[4], Clifford R. Jack[4], William Jagust[5], John Q. Trojanowski[6], Arthur W. Toga[7], Laurel Beckett[8], Robert C. Green[9], Andrew J. Saykin[10], John Morris[11], Leslie M. Shaw[6], Zaven Khachaturian[8,12], Greg Sorensen[13], Lew Kuller[14], Marcus Raichle[11], Steven Paul[15], Peter Davies[16], Howard Fillit[17], Franz Hefti[18], David Holtzman[11], Marek M. Mesulam[19], William Potter[20], Peter Snyder[21], Adam Schwartz[22], Tom Montine[23], Ronald G. Thomas[23], Michael Donohue[23], Sarah Walter[23], Devon Gessert[23], Tamie Sather[23], Gus Jiminez[23], Danielle Harvey[8], Matthew Bernstein[4], Paul Thompson[24], Norbert Schuff[2,8], Bret Borowski[4], Jeff Gunter[4], Matt Senjem[4], Prashanthi Vemuri[4], David Jones[4], Kejal Kantarci[4], Chad Ward[4], Robert A. Koeppe[25], Norm Foster[26], Eric M. Reiman[27], Kewei Chen[27], Chet Mathis[14], Susan Landau[5], Nigel J. Cairns[11], Erin Householder[11], Lisa Taylor-Reinwald[11], Virginia Lee[6], Magdalena Korecka[6], Michal Figurski[6], Karen Crawford[7], Scott Neu[7], Tatiana M. Foroud[10], Steven G. Potkin[28], Li Shen[10], Kelley Faber[10], Sungeun Kim[10], Kwangsik Nho[10], Leon Thal[3], Neil Buckholtz[29], Marylyn Albert[30], Richard Frank[31], John Hsiao[29], Jeffrey Kaye[32], Joseph Quinn[32], Betty Lind[32], Raina Carter[32], Sara Dolen[32], Lon S. Schneider[7], Sonia Pawluczyk[7], Mauricio Beccera[7], Liberty Teodoro[7], Bryan M. Spann[7], James Brewer[3], Helen Vanderswag[3], Adam Fleisher[3,27], Judith L. Heidebrink[25], Joanne L. Lord[25], Sara S. Mason[4], Colleen S. Albers[4], David Knopman[4], Kris Johnson[4], Rachelle S. Doody[33], Javier Villanueva-Meyer[33], Munir Chowdhury[33], Susan Rountree[33], Mimi Dang[33], Yaakov Stern[33], Lawrence S. Honig[33], Karen L. Bell[33], Beau Ances[11], Maria Carroll[11], Sue Leon[11], Mark A. Mintun[11], Stacy Schneider[11], Angela Oliver[11], Daniel Marson[34], Randall Griffith[34], David Clark[34], David Geldmacher[34], John Brockington[34], Erik Roberson[34], Hillel Grossman[35], Effie Mitsis[35], Leyla de Toledo-Morrell[36], Raj C. Shah[36], Ranjan Duara[37], Daniel Varon[37], Maria T. Greig[37], Peggy Roberts[37], Chiadi Onyike[30], Daniel D'Agostino[30], Stephanie Kielb[30], James E. Galvin[38], Brittany Cerbone[38], Christina A. Michel[38], Henry Rusinek[38], Mony J. de Leon[38], Lidia Glodzik[38], Susan De Santi[38], P Murali Doraiswamy[39], Jeffrey R. Petrella[39], Terence Z. Wong[39], Steven E. Arnold[6], Jason H. Karlawish[6], David Wolk[6], Charles D. Smith[40], Greg Jicha[40], Peter Hardy[40], Partha Sinha[40], Elizabeth Oates[40], Gary Conrad[40], Oscar L. Lopez[14], MaryAnn Oakley[14], Donna M. Simpson[30], Anton P. Porsteinsson[41], Bonnie S. Goldstein[41], Kim Martin[41], Kelly M. Makino[41], M Saleem Ismail[41], Connie Brand[41], Ruth A. Mulnard[28], Gaby Thai[28], Catherine McAdams-Ortiz[28], Kyle Womack[42], Dana Mathews[42], Mary Quiceno[42], Ramon Diaz-Arrastia[42], Richard King[42], Myron Weiner[42], Kristen Martin-Cook[42], Michael DeVous[42], Allan I Levey[43], James J. Lah[43], Janet S. Cellar[43], Jeffrey M. Burns[44], Heather S. Anderson[44], Russell H. Swerdlow[44], Liana Apostolova[24], Kathleen Tingus[24], Ellen Woo[24], Daniel H.S. Silverman[24], Po H. Lu[24], George Bartzokis[24], Neill R. Graff-Radford[45], Francine Parfitt[45], Tracy Kendall[45], Heather Johnson[45], Martin R. Farlow[10], Ann Marie Hake[10], Brandy R. Matthews[10],

Scott Herring[10], Cynthia Hunt[10], Christopher H. van Dyck[46], Richard E. Carson[46], Martha G. MacAvoy[46], Howard Chertkow[47], Howard Bergman[47], Chris Hosein[47], Ging-Yuek Robin Hsiung[48], Howard Feldman[48], Benita Mudge[48], Michele Assaly[48], Charles Bernick[49], Donna Munic[49], Andrew Kertesz[50], John Rogers[50], Dick Trost[50], Diana Kerwin[19], Kristine Lipowski[19], Chuang-Kuo Wu[19], Nancy Johnson[19], Carl Sadowsky[51], Walter Martinez[51], Teresa Villena[51], Raymond Scott Turner[52], Kathleen Johnson[52], Brigid Reynolds[52], Reisa A. Sperling[9], Keith A. Johnson[9], Gad Marshall[9], Meghan Frey[9], Barton Lane[9], Allyson Rosen[9], Jared Tinklenberg[9], Marwan N. Sabbagh[53], Christine M. Belden[53], Sandra A. Jacobson[53], Sherye A. Sirrel[53], Neil Kowall[53], Ronald Killiany[54], Andrew E. Budson[54], Alexander Norbash[54], Patricia Lynn Johnson[54], Joanne Allard[55], Alan Lerner[56], Paula Ogrocki[56], Leon Hudson[56], Evan Fletcher[8], Owen Carmichael[8], John Olichney[8], Charles DeCarli[8], Smita Kittur[57], Michael Borrie[58], T-Y. Lee[58], Rob Bartha[58], Sterling Johnson[59], Sanjay Asthana[59], Cynthia M. Carlsson[59], Adrian Preda[24], Dana Nguyen[24], Pierre Tariot[26], Stephanie Reeder[26], Vernice Bates[60], Horacio Capote[60], Michelle Rainka[60], Douglas W. Scharre[61], Maria Kataki[61], Anahita Adeli[61], Earl A. Zimmerman[62], Dzintra Celmins[62], Alice D. Brown[62], Godfrey D. Pearlson[63], Karen Blank[63], Karen Anderson[63], Robert B. Santulli[64], Tamar J. Kitzmiller[64], Eben S. Schwartz[64], Kaycee M. Sink[65], Jeff D. Williamson[65], Pradeep Garg[65], Franklin Watkins[65], Brian R. Ott[66], Henry Querfurth[66], Geoffrey Tremont[66], Stephen Salloway[67], Paul Malloy[67], Stephen Correia[67], Howard J. Rosen[2], Bruce L. Miller[2], Jacobo Mintzer[68], Kenneth Spicer[68], David Bachman[68], Stephen Pasternak[50], Irina Rachinsky[50], Dick Drost[50], Nunzio Pomara[69], Raymundo Hernando[69], Antero Sarrael[69], Susan K. Schultz[70], Laura L. Boles Ponto[70], Hyungsub Shim[70], Karen Elizabeth Smith[70], Norman Relkin[15], Gloria Chaing[15], Lisa Raudin[12,15], Amanda Smith[71], Kristin Fargher[71], Balebail Ashok Raj[71], Thomas Neylan[2], Jordan Grafman[19], Melissa Davis[3], Rosemary Morrison[3], Jacqueline Hayes[2], Shannon Finley[2], Karl Friedl[72], Debra Fleischman[36], Konstantinos Arfanakis[36], Olga James[39], Dino Massoglia[68], J Jay Fruehling[59], Sandra Harding[59], Elaine R. Peskind[23], Eric C. Petrie[61], Gail Li[61], Jerome A. Yesavage[73], Joy L. Taylor[73] & Ansgar J. Furst[73]

[2]UC San Francisco, San Francisco, CA 94143, USA. [3]UC San Diego, San Diego, CA 92093, USA. [4]Mayo Clinic, Rochester, NY 14603, USA. [5]UC Berkeley, Berkeley, CA 94720, USA. [6]UPenn, Philadelphia, PA 9104, USA. [7]USC, Los Angeles, CA 90089, USA. [8]UC Davis, Davis, CA 95616, USA. [9]Brigham and Women's Hospital/Harvard Medical School, Boston, MA 02115, USA. [10]Indiana University, Bloomington, IN 47405, USA. [11]Washington University in St Louis, St Louis, MI 63130, USA. [12]Prevent Alzheimer's Disease 2020, Rockville, MD 20850, USA. [13]Siemens, Munich 80333, Germany. [14]University of Pittsburgh, Pittsburgh, PA 15260, USA. [15]Weill Cornell Medical College, Cornell University, New York City, NY 10065, USA. [16]Albert Einstein College of Medicine of Yeshiva University, Bronx, NY 10461, USA. [17]AD Drug Discovery Foundation, New York City, NY 10019, USA. [18]Acumen Pharmaceuticals, Livermore, CA 94551, USA. [19]Northwestern University, Evanston and Chicago, IL 60208, USA. [20]National Institute of Mental Health, Rockville, MD 20852, USA. [21]Brown University, Providence, RI 02912, USA. [22]Eli Lilly, Indianapolis, IN 46225, USA. [23]University of Washington, Seattle, WA 98195, USA. [24]UCLA, Los Angeles, CA 90095, USA. [25]University of Michigan, Ann Arbor, MI 48109, USA. [26]University of Utah, Salt Lake City, UT 84112, USA. [27]Banner Alzheimer's Institute, Phoenix, AZ 85006, USA. [28]UC Irvine, Irvine, CA 92697, USA. [29]National Institute on Aging, Bethesda, MD 20892, USA. [30]Johns Hopkins University, Baltimore, MD 21218, USA. [31]Richard Frank Consulting, Washington, DC 20001, USA. [32]Oregon Health and Science University, Portland, OR 97239, USA. [33]Baylor College of Medicine, Houston, TX 77030, USA. [34]University of Alabama, Birmingham, AL 35233, USA. [35]Mount Sinai School of Medicine, New York City, NY 10029, USA. [36]Rush University Medical Center, Chicago, IL 60612, USA. [37]Wien Center, Miami, FL 33140, USA. [38]New York University, New York City, NY 10003, USA. [39]Duke University Medical Center, Durham, NC 27710, USA. [40]University of Kentucky, Lexington, KY 0506, USA. [41]University of Rochester Medical Center, Rochester, NY 14642, USA. [42]University of Texas Southwestern Medical School, Dallas, TX 75390, USA. [43]Emory University, Atlanta, GA 30322, USA. [44]Medical Center, University of Kansas, Kansas City, KS 66103, USA. [45]Mayo Clinic, Jacksonville, FL 32224, USA. [46]Yale University School of Medicine, New Haven, CT 06510, USA. [47]McGill University/Montreal-Jewish General Hospital, Montreal, QC H3T 1E2, Canada. [48]University of British Columbia Clinic for AD & Related Disorders, Vancouver, BC V6T 1Z3, Canada. [49]Cleveland Clinic Lou Ruvo Center for Brain Health, Las Vegas, NV 89106, USA. [50]St Joseph's Health Care, London, ON N6A 4V2, Canada. [51]Premiere Research Institute, Palm Beach Neurology, Miami, FL 33407, USA. [52]Georgetown University Medical Center, Washington, DC 20007, USA. [53]Banner Sun Health Research Institute, Sun City, AZ 85351, USA. [54]Boston University, Boston, MA 02215, USA. [55]Howard University, Washington, DC 20059, USA. [56]Case Western Reserve University, Cleveland, OH 20002, USA. [57]Neurological Care of CNY, Liverpool, NY 13088, USA. [58]Parkwood Hospital, London, ON N6C 0A7, Canada. [59]University of Wisconsin, Madison, WI 53706, USA. [60]Dent Neurologic Institute, Amherst, NY 14226, USA. [61]Ohio State University, Columbus, OH 43210, USA. [62]Albany Medical College, Albany, NY 12208, USA. [63]Hartford Hospital, Olin Neuropsychiatry Research Center, Hartford, CT 06114, USA. [64]Dartmouth- Hitchcock Medical Center, Lebanon, NH 03766, USA. [65]Wake Forest University Health Sciences, Winston-Salem, NC 27157, USA. [66]Rhode Island Hospital, Providence, RI 02903, USA. [67]Butler Hospital, Providence, RI 02906, USA. [68]Medical University South Carolina, Charleston, SC 29425, USA. [69]Nathan Kline Institute, Orangeburg, NY 10962, USA. [70]University of Iowa College of Medicine, Iowa City, IA 52242, USA. [71]University of South Florida: USF Health Byrd Alzheimer's Institute, Tampa, FL 33613, USA. [72]Department of Defense, Arlington, VA 22350, USA. [73]Stanford University, Stanford, CA 94305, USA

