## [Peer Review File · Nature Communications]

Reviewers' comments:

Reviewer #1 (Remarks to the Author):

The colleagues Franzmeier et al report here that the SNP rs744373 in BIN1 is associated with increased Tau pathology but not beta-amyloid pathology as measured respectively by AV1451 and AV45 PET in non-demented elderly.

This study represents an important addition to current knowledge described in previous studies, that is 1/ demonstration in vivo of the association of BIN1 risk allele with increased Tau pathology 2/ demonstration of such association in non-demented elderly 3/demonstration of such association in different affected brain regions. That important addition to current data needs to be better specified. More specifically in the introduction, reference 14 is a review that refers mainly to reference 17 when formulating the hypothesis that "the role of Bin1 in AD is the aggravation of tau pathology". In the manuscript referenced as 17 it was already demonstrated that "a Bin1 risk allele was associated with Tau pathology and not with A β 40 and A β 42 pathology in the brains of AD patients" and not only that the risk allele of BIN1 is associated with increased Bin1 mRNA levels as suggested in the introduction. In this study that was demonstrated ex-vivo on AD brain sections using AT-8 IHC as a proxy for Tau pathology (as done for the Braak staging). Again, this manuscript does add significant additional knowledge but is somewhat overstating novelty which can be easily corrected.

Note that in the discussion (line 269-270) it is mentioned that reference 17 describes that "BIN1 risk variants have been previously shown to correlate with post-mortem assessed brain tau levels but not with amyloid". This is not correct since reference 17 demonstrated AT8 positive Tau pathology in post mortem brain rather than tau levels to be correlated with a BIN1 risk variant.

The current study describes the association between the BIN1 risk allele and increased Tau pathology in cognitive normal and MCI subject which is an important addition to the current knowledge. It would however have been very valuable to have the previously report of such association in AD patients described on brain sections (ref 17) confirmed using a different technology. Can AD patients be included in the current study?

It is of interest to see that there is a possibly a spatial correlation between BIN1 mRNA levels and tau pathology. In light of previous reports on expression level changes depending on the BIN1 isoform it would be more relevant to analyze expression levels of different isoforms. As indicated in the discussion, this needs to be clarified. Can this be done based on the microarray data – can for example isoform specific probe sets be identified within the available microarray data? Can as such the dichotomy in expression changes between the neuron specific and other BIN1 isoforms be analysed?

Minor comment: the description that "region specificity was analyzed by studying tau PET SUVR within the 6 Braak stages" is confusing. The 6 Braak stages are referring to different spatial stages of Tau pathology (AT8 positivity) in the brain and not brain regions that are affected in such stages. In non-demented individuals higher Braak stages are very rare, hence this causes confusion with the readers.

Reviewer #2 (Remarks to the Author):

In this report, Franzmeier et al. describe the first in vivo evidence that a BIN1 polymorphism (rs744373) is associated with increased tau, but not, beta-amyloid pathology in live subjects. This provides direct patient evidence that altered BIN1 function might contribute to risk of dementia and AD via altered tau pathology. This is a well designed and executed study that provides vital support for a role of BIN1 in the development of a critical facet of AD pathology, which has long

been suspected but, until now, not entirely proven.

Reviewer #3 (Remarks to the Author):

This paper presents data from 89 individuals from the ADNI cohort showing that the BIN1 SNP, rs744373 was associated with increased tau uptake using functional brain imaging. Further they report that this same allele was associated with poorer memory performance.

Critique:

1. It has been established that brain expression of BIN1 is increased in Alzheimer's disease, thus the expression and imaging data here is incremental.
2. It is unclear whether someone had to first be amyloid positive to see the genotype effect on the amount of tau binding. The authors did not stratify by amyloid positive or negative status when investigating the association between BIN1 and tau.
3. Fig 3 is confusing. The authors performed a linear regression of the effects of AV-45 (amyloid) and BIN1 genotype on AV-1451 (tau) binding, but the figure shows correlation plots between AV-45 and AV-1451. They comment that for any given amount of amyloid, the amount of tau binding is greater in BIN1 carriers than non-carriers. But would a Pearson correlation between AV-1451 and AV-45 for either BIN1 carriers or non-carriers be statistically significant (looking at the raw data)? The slopes of the best fit lines are flat.
4. Table 1, suggests that the effect of BIN1 is entirely accounted for by the BIN1-risk MCI group, which are 2:1 amyloid + to amyloid negative. Therefore, the amyloid + MCI may be accounting for the association. These folks already have tau and cognitive impairment, and are at high risk of dementia regardless of BIN1 genotype.
5. While BIN1 appears to influence the amount of tau in amyloid + MCI patients, whether it really increases risk of dementia is unclear. I don't think it's been demonstrated that the amount of tau is as important a risk for dementia as whether it's present at all.
6. A recent paper below suggests that BIN1 localization is not in close proximity to neurofibrillary tangles.
7. The "second major finding" reported by this group is the association with poor memory, but the cohort included MCI. Whether the authors agree or not, these folks may already have AD explaining most of their findings.

De Rossi P et al. Aberrant accrual of BIN1 near Alzheimer's disease amyloid deposits in transgenic models. *Brain Pathol.* 2018 PubMed PMID: 30506549.

De Rossi P, et al. BIN1 localization is distinct from Tau tangles in Alzheimer's disease. *Matters (Zur).* 2017 Jan;2017. PubMed PMID: 29479533; PubMed Central PMCID: PMC5823513.

De Rossi P, et al. Predominant expression of Alzheimer's disease-associated BIN1 in mature oligodendrocytes and localization to white matter tracts. *Mol Neurodegener.* 2016;11 :59. PubMed PMID: 27488240; PubMed Central PMCID: PMC4973113.

Letter of response

Franzmeier et al. "The BIN1 rs744373 SNP is associated with increased tau-PET levels and worse memory.

Reviewer #1 (Remarks to the Author):

Reviewer: The colleagues Franzmeier et al report here that the SNP rs744373 in BIN1 is associated with increased Tau pathology but not beta-amyloid pathology as measured respectively by AV1451 and AV45 PET in non-demented elderly.

This study represents an important addition to current knowledge described in previous studies, that is 1/ demonstration in vivo of the association of BIN1 risk allele with increased Tau pathology 2/ demonstration of such association in non-demented elderly 3/demonstration of such association in different affected brain regions. That important addition to current data needs to be better specified.

Response: We appreciate these suggestions. We have adopted these points and made now clearer the motivation of our study and the importance of our findings (p.7 introduction, and p. 13 in discussion).

Reviewer: More specifically in the introduction, reference 14 is a review that refers mainly to reference 17 when formulating the hypothesis that "the role of Bin1 in AD is the aggravation of tau pathology". In the manuscript referenced as 17 it was already demonstrated that "a Bin1 risk allele was associated with Tau pathology and not with A β 40 and A β 42 pathology in the brains of AD patients" and not only that the risk allele of BIN1 is associated with increased Bin1 mRNA levels as suggested in the introduction. In this study that was demonstrated ex-vivo on AD brain sections using AT-8 IHC as a proxy for Tau pathology (as done for the Braak staging). Again, this manuscript does add significant additional knowledge but is somewhat overstating novelty which can be easily corrected.

Response: We thank the reviewer for this comment. We have corrected the referencing in the introduction. Specifically, we now state that "previous studies have described higher BIN1 mRNA expression in brain tissue of BIN1 risk SNP carriers ^{1,2}. Further, AD patients carrying a BIN1 risk SNP showed higher post-mortem tau pathology but not amyloid when compared to non-carrier AD patients ² (p.5).

Reviewer: Note that in the discussion (line 269-270) it is mentioned that reference 17 describes that "BIN1 risk variants have been previously shown to correlate with post-mortem assessed brain tau levels but not with amyloid". This is not correct since reference 17 demonstrated AT8 positive Tau pathology in post mortem brain rather than tau levels to be correlated with a BIN1 risk variant.

Response: We thank the reviewer for catching this mistake, we have corrected the relevant section, which now reads "...BIN1 risk variants have been previously shown to correlate with post-mortem assessed brain levels of AT8 positive tau pathology but not with A β (p.13).

Reviewer: The current study describes the association between the BIN1 risk allele and increased Tau pathology in cognitive normal and MCI subject which is an important addition

to the current knowledge. It would however have been very valuable to have the previously report of such association in AD patients described on brain sections (ref 17) confirmed using a different technology. Can AD patients be included in the current study?

Response: We agree that including patients with AD dementia would have been a valuable addition to the current data. However, for the current study, only 3 subjects with AD dementia met the inclusion criteria, of whom none carries the BIN1 rs744373 risk-allele (stated on p.18). We note however that according to the recent diagnostic frameworks for the biomarker-based diagnosis of AD, the MCI A β + subjects with evidence of tau pathology fulfilled the criteria of AD³. Thus, although no AD dementia patients were included, the current study does include subjects fulfilling research diagnostic criteria of AD.

Reviewer: It is of interest to see that there is a possibly a spatial correlation between BIN1 mRNA levels and tau pathology. In light of previous reports on expression level changes depending on the BIN1 isoform it would be more relevant to analyze expression levels of different isoforms. As indicated in the discussion, this needs to be clarified. Can this be done based on the microarray data – can for example isoform specific probe sets be identified within the available microarray data? Can as such the dichotomy in expression changes between the neuron specific and other BIN1 isoforms be analysed?

Response: We agree that analyzing the expression of BIN1 isoforms in relation to in-vivo tau pathology would be relevant. The currently used BIN1 mRNA expression data, however, were derived from the Allen brain atlas, which does not distinguish between different BIN1 isoforms. As such, the nature of the current data does not allow assessing any associations between isoform specific BIN1 expression and tau pathology. We acknowledge that BIN1 isoforms are differentially expressed in the brain and other body organs. However, within the brain, regional differences in isoform-specific expression levels other than a distinction between grey matter and white matter⁴ have not been yet established. Given that the Allen atlas includes brains from healthy subjects, it is unclear whether any brain region-specific isoform distribution is present in the brain in the first place. We agree with the reviewer that a BIN1 isoform mapping comparing different brain regions would be desirable, but this is beyond the scope of the current study.

Reviewer: Minor comment: the description that "region specificity was analyzed by studying tau PET SUVR within the 6 Braak stages" is confusing. The 6 Braak stages are referring to different spatial stages of Tau pathology (AT8 positivity) in the brain and not brain regions that are affected in such stages. In non-demented individuals higher Braak stages are very rare, hence this causes confusion with the readers.

Response: We thank the reviewer for picking this up. We have clarified our terminology and now refer to "brain regions corresponding to Braak stages" (Abstract, p.9 & Methods, p.19)

Reviewer #2 (Remarks to the Author):

Reviewer: In this report, Franzmeier et al. describe the first in vivo evidence that a BIN1 polymorphism (rs744373) is associated with increased tau, but not, beta-amyloid pathology in live subjects. This provides direct patient evidence that altered BIN1 function might contribute to risk of dementia and AD via altered tau pathology. This is a well designed and executed study that provides vital support for a role of BIN1 in the development of a critical facet of AD pathology, which has long been suspected but, until now, not entirely proven.

Response: We thank the reviewer for these encouraging remarks.

Reviewer #3 (Remarks to the Author):

This paper presents data from 89 individuals from the ADNI cohort showing that the BIN1 SNP, rs744373 was associated with increased tau uptake using functional brain imaging. Further they report that this same allele was associated with poorer memory performance.

Critique:

1. Reviewer: It has been established that brain expression of BIN1 is increased in Alzheimer's disease, thus the expression and imaging data here is incremental.

Response: We thank the reviewer for the opportunity to clarify the novelty of your study. Our primary aim was to test the association between carriage of the BIN1 rs744373 risk-allele and regional levels of molecular PET-assessed pathological tau in non-demented subjects in-vivo. The spatial match between tau PET uptake and a priori BIN1 mRNA expression maps was demonstrated in a secondary analysis, underscoring that BIN1 rs744373-related tau alterations overlap in regions where BIN1 expression is typically found in the brain. Our findings strongly suggest that the BIN1 rs744373 risk-allele is associated with higher tau pathology in predilection areas of tau pathology (i.e. Braak stage regions), which may mediate the association between BIN1 and memory impairment. The major contribution of the current study was thus to demonstrate in living non-demented subjects the link between the BIN1 rs744373 SNP and regional patterns of tau PET.

2. Reviewer: It is unclear whether someone had to first be amyloid positive to see the genotype effect on the amount of tau binding. The authors did not stratify by amyloid positive or negative status when investigating the association between BIN1 and tau.

Response: This is an important point. In our main analysis on the effect of BIN1 rs744373 on regional AV1451 tau-PET uptake, we controlled for age, gender, education, diagnosis, and ApoE4. In further models (see table 2, model 2), we specifically addressed the influence of A β by computing additional regression analyses, where we included continuous global AV45 amyloid-PET levels as an additional covariate. This approach allows assessing whether the BIN1 rs744373 SNP has a statistically significant effect on AV1451 tau-PET uptake independent of A β and other covariates. Here, all results remained virtually unchanged, that is we found that BIN1 rs744373 risk-allele carriers had higher AV1451 tau-PET uptake (globally & within Braak II-VI regions), independent of the level of A β . These results also remained consistent when controlling for A β -status (i.e. A β ⁺ vs. A β ⁻) instead of continuous AV45 amyloid-PET levels. Further, we found no interactions between continuous AV45 amyloid-PET or A β -status and BIN1 rs744373 on tau-PET uptake, suggesting that the association between BIN1 rs744373 and AV1451 tau-PET uptake is not different between A β -groups. Together, these results clearly suggest that the effect of BIN1 rs744373 on tau-PET uptake is not dependent on the levels of A β . Given the absence of the interaction between BIN1 and A β status (i.e. A β ⁺ vs. A β ⁻), a post-hoc regression analysis stratified by A β status is not indicated. We would further like to point the reviewer to the revised version of Figure 3, which shows that BIN1 risk-allele carriers show higher tau across different quartiles of global AV45 amyloid-PET (see also response to following comment).

3. Reviewer: Fig 3 is confusing. The authors performed a linear regression of the effects of AV-45 (amyloid) and BIN1 genotype on AV-1451 (tau) binding, but the figure shows correlation plots between AV-45 and AV-1451. They comment that for any given amount of amyloid, the amount of tau binding is greater in BIN1 carriers than non-carriers. But would a Pearson correlation between AV-1451 and AV-45 for either BIN1 carriers or non-carriers be statistically significant (looking at the raw data)? The slopes of the best fit lines are flat.

Response: We thank the reviewer for this comment. The purpose of Figure 3 was to demonstrate that the effect of the BIN1 rs744373 SNP on AV1451 tau-PET is constant across different levels of A β (i.e. in BIN1 rs744373 risk-allele carriers, tau is always elevated), rather than that the association between AV45 amyloid-PET and AV1451 tau PET is independent of BIN1 rs744373 SNP. Following the reviewer's suggestion, we have now modified Figure 3 so that the A β -independent effect of the BIN1 rs744373 SNP on AV1451 tau-PET can be better appreciated by the reader. Based on the regression analysis described above, we have now plotted the effect of BIN1 rs744373 on global AV1451 tau-PET (y-axis) stratified by global AV45 amyloid-PET quartiles. The plot shows that AV1451 tau-PET uptake is higher in BIN1 rs744373 risk-allele carriers compared to non-carriers regardless of the level of global AV45 amyloid-PET. Note that while the regression analysis included global AV45 amyloid PET as a continuous measure, we used quartiles of amyloid PET for Figure 3 in order to illustrate the BIN1 rs744373 SNP effect across a fairly fine-grained gradient of amyloid PET while keeping the subgroups of amyloid PET sufficiently large. We believe that this way of presenting our results helps to clarify that the effects of BIN1 rs744373 on tau are not modulated by AV45 amyloid PET levels. The relevant part of the results section has been modified (p.10-11).

Note that given the lack of the interaction between the BIN1 rs744373 SNP and AV45 amyloid-PET, a post-hoc analysis of simple main effects of either BIN1 rs744373 (at different levels of A β) or of AV45 amyloid-PET (at different levels of BIN1 rs744373) is not justified from a statistical point of view. Consequently, we did not conduct post-hoc regression analyses in subgroups defined by either AV45 amyloid-PET or BIN1 rs744373 SNP variants.

4. Reviewer: Table 1, suggests that the effect of BIN1 is entirely accounted for by the BIN1-risk MCI group, which are 2:1 amyloid + to amyloid negative. Therefore, the amyloid + MCI may be accounting for the association. These folks already have tau and cognitive impairment, and are at high risk of dementia regardless of BIN1 genotype.

Response: We agree that the higher proportion of A β ⁺ subjects in the MCI group is a potential confounding variable that may enhance the association between BIN1 rs744373 and AV1451 tau-PET. However, several findings suggest that our results are not driven by A β -status, continuous AV45 amyloid-PET levels or diagnosis. First, as already mentioned above, the effect of BIN1 rs744373 on AV1451 tau-PET remains significant when controlling for the diagnosis and A β -status or continuous AV45 amyloid-PET levels (see Figure 3 & table 2, model 2). Second, the interaction between BIN1 and A β -status was non-significant, suggesting that the effect of the BIN1 rs744373 SNP on tau PET was not dependent on A β -status (p.10-11). Third, visual inspection of Figure 3 suggests that the effect of the BIN1 rs744373 SNP was similar at different amyloid quartiles. Fourth, the frequency distribution of A β -status was not statistically different between BIN1 groups (chi-squared test, p=0.846, table 1). Together these results on the effect of BIN1 rs744373 on AV1451 tau-PET suggest that our findings are not driven by the MCI A β ⁺ group.

5. Reviewer: While BIN1 appears to influence the amount of tau in amyloid + MCI patients, whether it really increases risk of dementia is unclear. I don't think it's been demonstrated that the amount of tau is as important a risk for dementia as whether its present at all.

Response: Previous biomarker studies using CSF or neuroimaging biomarkers of tau pathology have shown that higher levels of tau are associated with higher memory impairment, faster rates of cognitive decline⁵⁻⁷ and faster progression towards dementia⁸. In fact, higher levels of pathological tau are among the strongest predictors of risk of dementia in comparison to other standard biomarkers of AD⁹. It should be noted that tau pathology is a marker of disease progression, which is gradual in AD⁷. Together these results suggest that higher levels of tau pathology are associated in a linear fashion to more severe cognitive decline.

We like to clarify that we tested whether the BIN1 rs744373 SNP is associated with memory impairment rather than risk of AD dementia. Our mediation analysis confirmed that the effect of the BIN1 risk variant on memory performance was entirely mediated by tau PET levels. We note that these results strongly suggest that the BIN1 rs744373 SNP association with lower cognitive performance is related to alterations in AV1451-assessed tau pathology. We agree with the reviewer however that no conclusions can be made with regard to the risk of the clinical manifestation of AD dementia. To avoid any misunderstandings, we rephrased our conclusions more carefully where pertinent (Abstract, Results p.12, Discussion p.13& 15)

6. Reviewer: A recent paper below suggest that BIN1 localization is not in close proximity to neurofibrillary tangles.

Response: De Rossi et al.¹⁰ showed that the BIN1 protein does not colocalize with tau tangles at a microscopic resolution in brain slices of AD patients. This result is consistent with the results reported by Chapuis et al.². These authors showed that even though post-mortem analyzed BIN1 expression in the brain of AD patients was not colocalized exactly with neurofibrillary tangles, the BIN1 SNP was associated with higher AT8-detectable pathology in the same brains from AD patients. The authors concluded that BIN1 risk is associated with pre-tangle pathological tau². The exact mechanisms by which BIN1 is associated with tau pathology are not clear. Nevertheless, several lines of research suggest a link between BIN1 and tau pathology. The BIN1 protein is known to be involved in vesicle processing and located near the cell membrane¹¹. A previous study¹² showed that BIN1 modulates the vesicle mediated endocytosis of pathological tau, thereby increasing tau seeding. De Rossi et al. showed that BIN1 is most highly expressed within the white matter⁴. Given that the BIN1 protein interacts with physiological tau, a constituent of microtubules², it is possible that alterations of BIN1 in the white matter increase the development of pathological tau, which manifest distantly as neurofibrillary tangles in the soma. Thus, BIN1 alterations may also have distant effects. However, the exact mechanism need to be investigated in future studies. We have modified our discussion to address these points (p.16)

We caution that our results do not necessarily imply that BIN1 does not relate to levels of amyloid pathology. Although our analysis showed that the BIN1 rs744373 risk-allele is not associated with AV45 amyloid-PET levels consistent with previous post-mortem studies^{2,13}, a recent study showed that BIN1 accumulates in the vicinity of amyloid plaques in brain sections of AD patients and in an AD mouse model¹⁴. These results support the view that BIN1 is associated with the development of fibrillary amyloid pathology, but it remains to be tested whether the BIN1 rs744373 SNP is associated with higher BIN1 levels surrounding

amyloid plaques, which would support the notion that alterations in BIN1 may enhance amyloid plaques. (added in Discussion, p.13-14)

7. Reviewer: The “second major finding” reported by this group is the association with poor memory, but the cohort included MCI. Whether the authors agree or not, these folks may already have AD explaining most of their findings.

Response: We took several measures to ensure that diagnostic status was not a confounding factor. Importantly, the BIN1 rs744373 SNP genotypes were not more frequently distributed in the MCI subjects compared to HC or in the A β + compared to A β - subgroups. Our mediation analysis showed that the association between the BIN1 rs744373 SNP and lower memory performance is mediated by differences in tau PET levels. The mediation analysis was controlled for continuous levels of global AV45 amyloid-PET and diagnosis among other covariates, rendering it unlikely that global clinical severity contributed to the mediation effect. Together, this analysis suggests that the risk for AD dementia in BIN1 rs744373 SNP carriers as reported by GWAS is mediated by elevated tau pathology, that is we show a link between a risk gene and elevated AD pathology. We acknowledge that a future longitudinal study will be of interest in order to test whether BIN1 is associated with faster rates of tau accumulation and memory decline. We mention this perspective in the discussion (p.17).

References:

- 1 Bungenberg, J. *et al.* Gene expression variance in hippocampal tissue of temporal lobe epilepsy patients corresponds to differential memory performance. *Neurobiol Dis* **86**, 121-130, doi:10.1016/j.nbd.2015.11.011 (2016).
- 2 Chapuis, J. *et al.* Increased expression of BIN1 mediates Alzheimer genetic risk by modulating tau pathology. *Mol Psychiatry* **18**, 1225-1234, doi:10.1038/mp.2013.1 (2013).
- 3 Jack, C. R., Jr. *et al.* NIA-AA Research Framework: Toward a biological definition of Alzheimer's disease. *Alzheimers Dement* **14**, 535-562, doi:10.1016/j.jalz.2018.02.018 (2018).
- 4 De Rossi, P. *et al.* Predominant expression of Alzheimer's disease-associated BIN1 in mature oligodendrocytes and localization to white matter tracts. *Mol Neurodegener* **11**, 59, doi:10.1186/s13024-016-0124-1 (2016).
- 5 Scholl, M. *et al.* PET Imaging of Tau Deposition in the Aging Human Brain. *Neuron* **89**, 971-982, doi:10.1016/j.neuron.2016.01.028 (2016).
- 6 Maass, A. *et al.* Entorhinal Tau Pathology, Episodic Memory Decline, and Neurodegeneration in Aging. *The Journal of neuroscience : the official journal of the Society for Neuroscience* **38**, 530-543, doi:10.1523/JNEUROSCI.2028-17.2017 (2018).
- 7 Mattsson, N. *et al.* (18)F-AV-1451 and CSF T-tau and P-tau as biomarkers in Alzheimer's disease. *EMBO Mol Med* **9**, 1212-1223, doi:10.15252/emmm.201707809 (2017).
- 8 Ewers, M. *et al.* Multicenter assessment of CSF-phosphorylated tau for the prediction of conversion of MCI. *Neurology* **69**, 2205-2212, doi:10.1212/01.wnl.0000286944.22262.ff (2007).
- 9 Frolich, L. *et al.* Incremental value of biomarker combinations to predict progression of mild cognitive impairment to Alzheimer's dementia. *Alzheimers Res Ther* **9**, 84, doi:10.1186/s13195-017-0301-7 (2017).

- 10 De Rossi, P. *et al.* BIN1 localization is distinct from Tau tangles in Alzheimer's disease. *Matters (Zur)* **2017**, doi:10.19185/matters.201611000018 (2017).
- 11 Butler, M. H. *et al.* Amphiphysin II (SH3P9; BIN1), a member of the amphiphysin/Rvs family, is concentrated in the cortical cytomatrix of axon initial segments and nodes of ranvier in brain and around T tubules in skeletal muscle. *J Cell Biol* **137**, 1355-1367 (1997).
- 12 Calafate, S., Flavin, W., Verstreken, P. & Moechars, D. Loss of Bin1 Promotes the Propagation of Tau Pathology. *Cell Rep* **17**, 931-940, doi:10.1016/j.celrep.2016.09.063 (2016).
- 13 Holler, C. J. *et al.* Bridging integrator 1 (BIN1) protein expression increases in the Alzheimer's disease brain and correlates with neurofibrillary tangle pathology. *Journal of Alzheimer's disease : JAD* **42**, 1221-1227, doi:10.3233/JAD-132450 (2014).
- 14 De Rossi, P. *et al.* Aberrant accrual of BIN1 near Alzheimer's disease amyloid deposits in transgenic models. *Brain Pathol*, doi:10.1111/bpa.12687 (2018).

REVIEWERS' COMMENTS:

Reviewer #1 (Remarks to the Author):

We thank the authors for their reply and are looking forward to see this manuscript now published.

Reviewer #2 (Remarks to the Author):

The authors have responded well to prior critiques. No further concerns.

Reviewer #3 (Remarks to the Author):

The authors have done an excellent job of responding to my critiques. While I have no further issues to address, one has to wonder if the mutations in BIN1 are specific to tau in Alzheimer's disease only or would the association be observed in other tauopathies such as Frontotemporal Lobar Degeneration or Progressive Supranuclear Palsy. Certainly others will ask how specific this association is to AD.

REVIEWER COMMENTS:

We thank all reviewers for their comments.

Reviewer #3 (Remarks to the Author): The authors have done an excellent job of responding to my critiques. While I have no further issues to address, one has to wonder if the mutations in BIN1 are specific to tau in Alzheimer's disease only or would the association be observed in other tauopathies such as Frontotemporal Lobar Degeneration or Progressive Supranuclear Palsy. Certainly others will ask how specific this association is to AD.

Response: The reviewer raises an interesting point. To our best knowledge, only a single study investigated the association specifically between BIN1 SNP and the risk of a tauopathy other than AD, reporting no association between the BIN1 744373 and fronto-temporal dementia (FTD)¹. However, the sample size was low (including 530 FTD cases and 926) and included different variants of FTD. Thus, the results remain inconclusive. Given that scarce evidence of an association between BIN1 and tauopathies other than AD, it is likely that the genetic risk of BIN1 SNPs is specific to AD. However, this remains speculative at this point and needs to be investigated in future studies. We have added these remarks to the discussion (p. 17).

REFERENCES:

- 1 Ferrari, R. *et al.* A genome-wide screening and SNPs-to-genes approach to identify novel genetic risk factors associated with frontotemporal dementia. *Neurobiol Aging* **36**, 2904 e2913-2926, doi:10.1016/j.neurobiolaging.2015.06.005 (2015).